# Do You Keep an Eye on What I Ask? Mitigating Multimodal Hallucination via Attention-Guided Ensemble Decoding

**Yeongjae Cho** [1*], **Keonwoo Kim** [2*], **Taebaek Hwang** [3], **Sungzoon Cho** [1†]
[1]Seoul National University, [2]Kim & Chang AI&IT System Center, [3]Waddle Corporation
`yjcho@bdai.snu.ac.kr, gunny1254@gmail.com`
`taebaek@waddlelab.com, zoon@snu.ac.kr`

## Abstract

Recent advancements in Large Vision-Language Models (LVLMs) have significantly expanded their utility in tasks like image captioning and visual question answering. However, they still struggle with object hallucination, where models generate descriptions that inaccurately reflect the visual content by including nonexistent objects or misrepresenting existing ones. While previous methods, such as data augmentation and training-free approaches, strive to tackle this issue, they still encounter scalability challenges and often depend on additional external modules. In this work, we propose **Ensemble Decoding (ED)**, a novel strategy that splits the input image into sub-images and combines logit distributions by assigning weights through the attention map. Furthermore, we introduce ED adaptive plausibility constraint to calibrate logit distribution and FastED, a variant designed for speed-critical applications. Extensive experiments across hallucination benchmarks demonstrate that our proposed method achieves state-of-the-art performance, validating the effectiveness of our approach.

## 1 Introduction

Recent advancements in Large Language Models (LLMs) (Brown, 2020; Touvron et al., 2023a;b; Jiang et al., 2023) have extended their capabilities into the visual domain. In particular, Large Vision-Language Models (LVLMs) (Liu et al., 2023b; Bai et al., 2023; Liu et al., 2024b; Dai et al., 2023; Gong et al., 2023) process visual inputs and generate contextually relevant text, making them effective for tasks such as image captioning and visual question answering. Despite extensive research focusing on optimizing LVLM architectures, training paradigms, and dataset combinations, the persistent issue of object hallucination raises significant concerns about the reliability and applicability of these models (Liu et al., 2023a; Lovenia et al., 2023; Li et al., 2023b; Liu et al., 2024a). Object hallucination occurs when LVLMs inaccurately describe visual content, misrepresenting or introducing nonexistent objects.

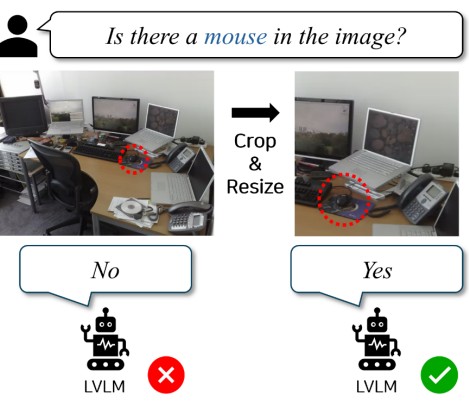

Figure 1: Example of object hallucination in LVLM (left). After cropping and resizing the image, the model answers correctly (right).

Object hallucination is especially problematic in applications that demand precise answers, such as autonomous vehicles (Iberraken & Adouane, 2023) and manufacturing systems (Mohammadi Amin et al., 2020). To address the issue, researchers have proposed strategies such as data augmentation and fine-tuning with specific datasets (Rohrbach et al., 2018; Gunjal et al., 2024), but these

---

*Equal contribution.
†Corresponding author.

approaches often struggle with scalability and generalization. Recently, training-free methods have emerged, including contrasting logit differences across layers (Chuang et al., 2023), retrospection-reallocation using logit penalties (Huang et al., 2024), contrastive decoding with noise (Leng et al., 2024), and the combination of local and global attention (An et al., 2024). While these methods show substantial improvement, they often fail to exploit intrinsic visual information fully.

As illustrated in Figure 1, we observe that inputting a relevant sub-image, a portion of the original image, into the model improves the performance of the generated response. Motivated by this observation, we conduct a pilot study to analyze how visual characteristics influence the model's visual representation. Our analysis identifies two key factors that can negatively affect model performance: (1) unnecessary objects in the image, and (2) low object resolution within the image. First, images containing numerous irrelevant objects to the question disrupt the model's focus, leading to object hallucinations. Second, when the object resolution in the image is low, it impairs the model's ability to interpret visual content accurately, resulting in incorrect inferences. This analysis suggests that instead of spreading focus across all areas, concentrating on selected sub-images enhances the model's ability to mitigate object hallucinations by effectively harnessing intrinsic visual information.

Motivated by these findings, we introduce Ensemble Decoding (ED), a novel training-free decoding strategy for LVLMs to mitigate object hallucination. ED splits the input image into sub-images by dividing the original image into several parts. It then aggregates logit distributions of each sub-image, which contain fewer objects and a higher resolution per object, along with those of the original image. ED leverages attention maps to prioritize sub-images, dynamically assigning different weights at each step of token generation to emphasize the necessary parts at that specific moment. This approach addresses the limitations of existing methods by adaptively exploiting intrinsic visual information without relying on additional modules. Furthermore, to consolidate the ensembled logits from sub-images more effectively, we introduce the ED adaptive plausibility constraint, which calibrates logit distributions to ensure fine-grained tokens contribute to the output. Additionally, since processing multiple sub-images increases the computational cost of the ED process, we develop FastED, an optimized variant that balances performance and speed by referencing only the sub-image with the highest mean attention score. Through extensive experiments on object hallucination benchmarks, we demonstrate that our proposed method achieves state-of-the-art results across most benchmark evaluation metrics, outperforming existing methods and confirming the effectiveness of our approach. The contributions of this paper are fourfold:

- We propose ED, a training-free decoding strategy for LVLMs that mitigates object hallucination by leveraging attention maps to split the input image into sub-images and combining their logit distributions.

- We introduce the ED adaptive plausibility constraint, which calibrates logits across multiple images to ensure fine-grained tokens contribute to the output.

- We develop FastED, an optimized variant of ED, balancing performance with speed by selecting a sub-image with the highest mean attention score from the original image.

- Through extensive experiments, we demonstrate that ED achieves state-of-the-art results across most benchmark evaluation metrics, outperforming existing methods.

## 2 Pilot Study

In this study, we investigate whether properly divided sub-images can reduce object hallucination in the outputs of LVLMs. To assess whether the LVLMs focus on the object relevant to the query while generating tokens, we employ attention maps. Cha et al. (2024) highlights that patch-wise projectors (Liu et al., 2023b; Chen et al., 2023a) preserve spatial locality, enabling more precise attention maps compared with resampler-based models (Bai et al., 2023; Dai et al., 2023; Ye et al., 2023; Zhu et al., 2023). Therefore, we adopt patch-wise projectors, specifically LLaVA-1.5 (Liu et al., 2024b), as the base model. We hypothesize that irrelevant objects and low object resolution in images are likely to impact performance negatively. To validate this hypothesis, we conduct experiments using grid-arranged images with randomly placed objects, accompanied by questions about specific objects in the image. Our objective is to verify if the model's highest mean attention score consistently aligns with the correct grid cell before generating an answer.

In our experimental setup, we manipulate two variables: (1) masking 1/4 of the unnecessary area to reduce irrelevant objects and (2) lowering the object resolution in the image, in order to assess their impact on model performance. As shown in Figure 2, applying masking to unnecessary objects improves performance, while lowering the resolution of objects leads to a decline. This trend becomes more pronounced as the number of objects increases. As sub-images generally contain fewer unnecessary objects and maintain a relatively higher resolution for each object compared to the original images, our findings confirm that sub-images help the model better attend to relevant objects. This is crucial for mitigating object hallucinations and improving overall performance (Yang et al., 2021), thus supporting the use of sub-images in our method. Further details are available in Appendix A.

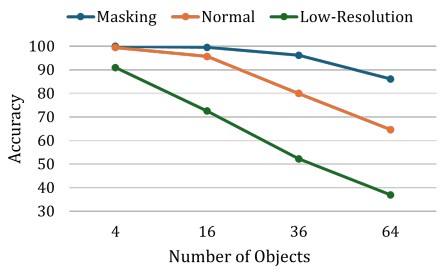

Figure 2: Experimental results of the pilot study. *Masking* refers to masking some irrelevant objects in the image, while *Low-Resolution* involves reducing the resolution of each object in the image.

## 3 METHOD

**Overview**   We present the input structure and notation for our proposed Ensemble Decoding (ED), as illustrated in Figure 3. In this approach, raw input images are split into multiple sub-images, which, along with the original raw image, are fed into a pre-trained LVLM, denoted as $p_\theta$ parameterized by $\theta$. Given a raw input image $v$, we split it into $N$ sub-images, each of size $c \times c$, denoted as $v_1, v_2, \ldots, v_N \in \mathcal{R}^{c \times c}$. The original image $v$ and the $N$ sub-images, along with the text $x$, are separately fed into the LVLM, resulting in a total of $N + 1$ image inputs.

### 3.1 ENSEMBLE DECODING

**Attention-Guided Weight**   In Ensemble Decoding, we calculate attention-guided weights for multiple sub-images derived from the raw input image at each decoding step $t$. The following equations detail the steps involved in the process. First of all, we compute the attention matrix $A$ at time step $t$ using the query $Q_t$ and key $K_t$ matrices from the self-attention mechanism (Vaswani, 2017), which involves the text $x$, the image $v$, and the previously generated tokens $y_{<t}$ :

$$A_t = \text{softmax}\left(\frac{Q_t K_t^T}{\sqrt{d_k}}\right), \tag{1}$$

where $d_k$ denotes the dimension of the key vectors. Following a similar approach to Lee et al. (2023), we select the top $K$ layers with the highest mean attention scores and average them to form a single representative layer to improve the attention matrix. Within the multi-head attention mechanism, we identify the top $H$ heads with the highest mean attention scores and average them to obtain a single attention matrix. The resulting attention matrix $\hat{A}_t$ is reshaped into a matrix of size $d \times d$:

$$\hat{A}_t = \frac{1}{H} \sum_{j=1}^{H} \text{sorted}\left(\frac{1}{K} \sum_{i=1}^{K} \text{sorted}(A_t)[i]\right)[j] \in \mathbb{R}^{d \times d}, \tag{2}$$

where $d \times d$ denotes the number of patches. Subsequently, we aggregate the refined attention matrix corresponding to each of the $N$ sub-images. Specifically, we identify the regions in $\hat{A}_t$ that correspond to each sub-image and sum the attention values within these regions to obtain the aggregated attention scores $s_{k,t}$, where $k$ indicates the index of each sub-image:

$$s_{k,t} = \sum_{(i,j) \in \text{Region}_k} \hat{A}_{t,i,j} \quad \text{for} \quad k = 1, 2, \ldots, N. \tag{3}$$

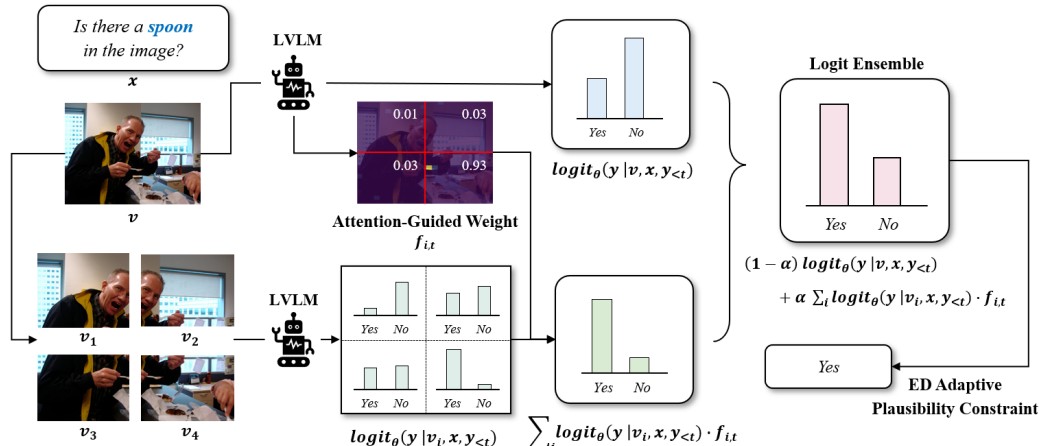

Figure 3: Overall pipeline of Ensemble Decoding (ED). Attention-guided weights are applied to sub-images and combined with the logits from the original image for ensembling. ED adaptive plausibility constraint is applied to generate the final output. The entire process is dynamically repeated at each time step $t$ of token generation.

Finally, we convert the aggregated attention score into attention-guided weight by applying a softmax function with temperature $\tau$.

$$f_{k,t} = \frac{\exp(s_{k,t}/\tau)}{\sum_{j=1}^{N} \exp(s_{j,t}/\tau)}. \tag{4}$$

This assigns an attention-guided weight $f_{k,t}$ to each of the $N$ sub-images at each decoding step $t$, indicating its relative importance and ensuring that each sub-image contributes to the final output based on its significance.

**Logit Ensemble**  Each sub-image $v_k$ is processed along with the associated text $x$ and the previously generated tokens $y_{<t}$ to produce a logit distribution $\mathrm{logit}_\theta(y_t \mid v_k, x, y_{<t})$, where $y_t$ denotes the token at decoding step $t$. These logits are weighted by their corresponding attention-guided weight $f_{k,t}$. The final logit distribution is computed by combining the weighted sum of the sub-image logits with the logit distribution from the raw image $v$, using a weighted term $\alpha \in [0, 1]$

$$p_{\mathrm{ED}}(y_t \mid V, x, y_{<t}) = \mathrm{softmax}\left[(1-\alpha)\mathrm{logit}_\theta(y_t \mid v, x, y_{<t}) + \alpha \sum_{k=1}^{N} \mathrm{logit}_\theta(y_t \mid v_k, x, y_{<t}) \cdot f_{k,t}\right], \tag{5}$$

where $V$ denotes $v, v_{1:N}$ the set containing the raw image and sub-images. This approach uses both the global context of the raw image and the local context of the sub-images at each decoding step $t$.

## 3.2 ED ADAPTIVE PLAUSIBILITY CONSTRAINT

In previous studies, adaptive plausibility constraint has been utilized in single-image scenarios to preserve tokens with high original probabilities and discard less likely ones (Li et al., 2022; Leng et al., 2024; An et al., 2024). However, in ED, integrating probabilities from multiple images yields more detailed and fine-grained representations that must be preserved to ensure accurate analysis. To address this, we introduce the ED adaptive plausibility constraint. It adjusts the truncation strength through a hyperparameter $\beta$, which is applied to the weighted sum of probabilities calculated from each of the $N$ sub-images. The constraint is defined as follows:

$$\mathcal{V}_{\mathrm{head}}(y_{<t}) = \left\{ y_t \in \mathcal{V} : \sum_{k=1}^{N} p_\theta(y_t \mid v_k, x, y_{<t}) \geq \beta \max_w \left( \sum_{k=1}^{N} p_\theta(w \mid v_k, x, y_{<t}) \cdot f_{k,t} \right) \right\}. \tag{6}$$

The probability for tokens not in $\mathcal{V}_{\text{head}}(y_{<t})$ is set to zero:

$$p_{\text{ED}}(y_t \mid V, x) = 0, \quad \text{if} \quad y_t \notin \mathcal{V}_{\text{head}}(y_{<t}).$$

It ensures that only the most plausible tokens based on the weighted sum of logits across all sub-images contribute to the final probability distribution, preserving the integrity of ED.

### 3.3 FASTED

While ED enhances model performance by capturing finer details through multiple sub-images, it increases computational cost due to multiple forward passes. To balance performance and speed, we introduce FastED, which only uses the raw image and the sub-image with the highest attention-guided weight to compute the logit distribution. It significantly reduces the computation from $N + 1$ to 2 forward passes, with minimal performance trade-off, as demonstrated in our experimental section. The modified equation for FastED is as follows:

$$p_{\text{FastED}}(y_t \mid v, v_{k^*}, x, y_{<t}) = \text{softmax}\left[(1 - \alpha)\text{logit}_\theta(y_t \mid v, x, y_{<t}) + \alpha \cdot \text{logit}_\theta(y_t \mid v_{k^*}, x, y_{<t})\right], \tag{7}$$

where $k^*$ corresponds to the sub-image with the highest attention-guided weight $f_{k^*}$.

## 4 EXPERIMENTS

### 4.1 EXPERIMENTAL SETTINGS

**Dataset and Evaluation Criteria**   We evaluate ED on the POPE and CHAIR benchmarks, which focus on object existence hallucination, and further assess its performance on the MME and LLaVA-Bench, which evaluate additional attributes beyond object existence. Detailed descriptions of each dataset are provided below.

**POPE** (Polling-based Object Probing Evaluation) (Li et al., 2023b) includes 27,000 Yes/No questions across three datasets (MSCOCO, A-OKVQA, GQA) (Lin et al., 2014; Schwenk et al., 2022; Hudson & Manning, 2019), balanced equally between existent and non-existent objects. Non-existent samples are constructed using random, popular, and adversarial settings. We use precision, recall, F1 score, and accuracy as evaluation metrics.

**CHAIR** (Caption Hallucination Assessment with Image Relevance) (Rohrbach et al., 2018) measures object hallucination in image captions by calculating the proportion of objects mentioned that do not appear in the ground-truth labels. Following An et al. (2024), we randomly select images from the MSCOCO (Lin et al., 2014) and use CHAIR$_s$, CHAIR$_I$, and recall as evaluation metrics.

**MME** (Fu et al., 2024) provides a benchmark for assessing LVLMs across various tasks. Following Yin et al. (2023), we evaluate hallucination using four subtasks: *Existence*, *Count*, *Position*, and *Color*, with performance measured by the combined metric of accuracy and accuracy+.

**LLaVA-Bench** (Liu et al., 2023b) consists of 24 images and 60 questions to assess LVLMs' performance on complex tasks and domain adaptation. Following Leng et al. (2024), we use GPT-4 to assess the accuracy and detailedness of LVLM's image captioning using a 10-point Likert scale.

**Baselines**   As baselines, we use the commonly adopted multinomial sampling decoding method (Regular) and four other state-of-the-art training-free decoding strategies for object hallucination tasks. VCD (Leng et al., 2024) reduces hallucinations through contrastive decoding using noisy images. DoLA (Chuang et al., 2023) suppresses hallucinations by contrasting logits from the first and last layers of the model. OPERA (Huang et al., 2024) mitigates hallucinations by penalizing certain knowledge aggregation patterns. AGLA (An et al., 2024) tackles attention deficiency issues by integrating global attention with local attention derived from masked images.

**Implementation Details**   In all experiments applying ED, we set $N = 4$ to divide the input image into sub-images. The LVLM model used in all experiments is LLaVA-1.5 (Liu et al., 2024b). The

Table 1: Experimental results of POPE on different decoding strategies.

| Setting | Decoding | Precision | Recall | F1 Score | Accuracy |
|---|---|---|---|---|---|
| *Random* | Regular | 88.84 | 76.76 | 82.28 | 83.49 |
| | DOLA | 87.59 | 81.27 | 84.19 | 84.78 |
| | OPERA | 94.52 | 79.80 | 86.45 | 87.53 |
| | VCD | 87.15 | 86.68 | 86.83 | 86.84 |
| | AGLA | 94.41 | 82.08 | 87.71 | 88.54 |
| | **ED** | 93.40 | 86.41 | **89.68** | **90.08** |
| *Popular* | Regular | 82.47 | 76.76 | 79.34 | 79.98 |
| | DOLA | 84.11 | 76.22 | 80.61 | 79.75 |
| | OPERA | 88.00 | 79.80 | 83.50 | 84.21 |
| | VCD | 87.15 | 80.59 | 83.37 | 82.65 |
| | AGLA | 87.88 | 82.08 | 84.68 | 85.14 |
| | **ED** | 86.12 | 86.41 | **86.00** | **86.09** |
| *Adversarial* | Regular | 76.11 | 76.80 | 76.26 | 76.03 |
| | DOLA | 77.27 | 75.47 | 76.16 | 76.32 |
| | OPERA | 82.16 | 79.76 | 80.69 | 80.88 |
| | VCD | 73.43 | 86.47 | 79.28 | 77.31 |
| | AGLA | 81.20 | 82.10 | 81.36 | 81.13 |
| | **ED** | 79.75 | 86.47 | **81.90** | **82.75** |

Table 2: Experimental results on a subset of CHAIR with different decoding strategies. To ensure a fair comparison, we include the average length of generated outputs, as CHAIR metrics and recall can vary with different output lengths. Baseline results are referenced from An et al. (2024).

| Decoding | CHAIR$_s\downarrow$ | CHAIR$_I\downarrow$ | Recall$\uparrow$ | Average Length |
|---|---|---|---|---|
| Regular | 51.0 | 15.2 | 75.2 | 102.2 |
| DOLA | 57.0 | 15.9 | 78.2 | 97.5 |
| OPERA | 47.0 | 14.6 | 78.5 | 95.3 |
| VCD | 51.0 | 14.9 | 77.2 | 101.9 |
| AGLA | **43.0** | 14.1 | 78.9 | 98.8 |
| **ED** | **43.0** | **14.0** | **82.5** | 100.1 |

hyperparameters $\alpha$ and $\beta$ are set to 0.5. The softmax temperature $\tau$ is set to 1e-2 for short-answer tasks like POPE and MME, and 1e-4 for longer-answer tasks like CHAIR and LLaVA-Bench. The attention map generation follows (Lee et al., 2023), with $H$ and $K$ set to 3. We use an A6000 GPU, and all experiments are repeated three times, with reported results averaged.

## 4.2 MAIN RESULTS

In the main experiment, we evaluate the performance of ED on the POPE and CHAIR benchmarks, which assess the existence of objects. Table 1 presents the results of POPE in the *Random*, *Popular*, and *Adversarial* settings, where the ratio of Yes/No labels is 50%. In terms of F1 score and accuracy, ED shows an average improvement of 6.57%p and 6.47%p, respectively, compared to Regular, demonstrating the effectiveness of ED. Moreover, ED consistently outperforms previous state-of-the-art methods across all POPE settings.

Unlike POPE, which uses Yes/No labels to assess the presence of objects, Table 2 presents the results for the CHAIR benchmark which specifically evaluates the presence of object hallucinations in detailed image captioning. In this experiment, ED achieves the highest recall, indicating a significant improvement in the detailedness of generated captions. Compared to Regular, ED shows substantial

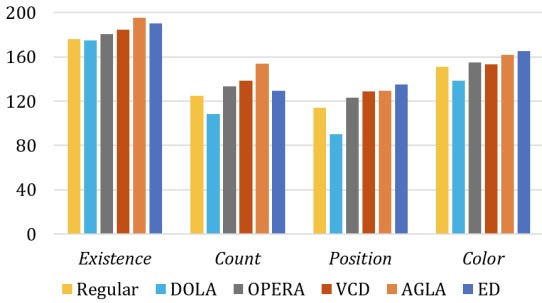

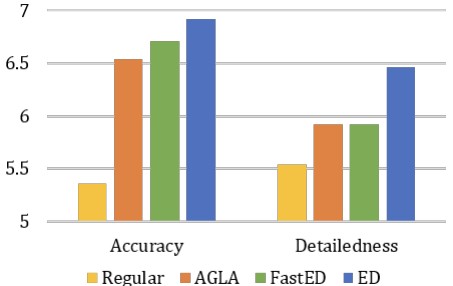

Figure 4: Experimental results of MME on a hallucination subset with different decoding strategies.

Figure 5: Results of GPT-aided evaluation on the captions of LLaVA-Bench.

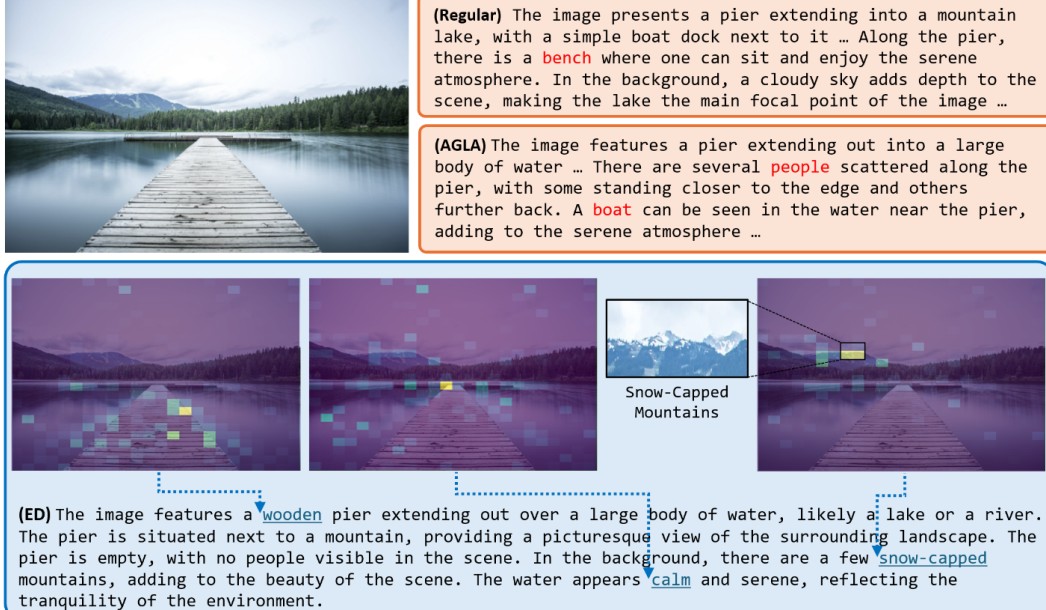

Figure 6: Generated captions using Regular, AGLA, and ED decoding strategies. Red text indicates hallucinations. Attention maps illustrate the state before generating each word.

performance improvements across all metrics. Specifically, ED outperforms the previous state-of-the-art method, AGLA, with a 3.6%p improvement in recall and a 0.1%p reduction in CHAIR$_I$↓. Although ED performs equally to AGLA on CHAIR$_S$↓, these results confirm ED's effectiveness in reducing object hallucinations in generated captions.

## 5 DISCUSSION

### 5.1 BEYOND OBJECT HALLUCINATION

In this section, we extend our analysis to include hallucinations involving object attributes. As shown in Figure 4, ED shows the highest performance on position and color-related questions and achieves near-perfect accuracy on existence-related questions on the hallucination subset of the MME dataset. However, it struggles with object counting, likely due to the image-splitting process fragmenting objects, which complicates quantity-related tasks. Moreover, we also evaluate the accuracy and detail level of captions generated on LLaVA-Bench using GPT-4. As detailed in Figure 5, both FastED and ED outperform Regular and the previous state-of-the-art model AGLA. ED, in particular, shows a significant improvement in detail compared to FastED. This performance boost is likely due to referencing all split images, rather than only the one with the highest mean attention score. By

leveraging the full image context, ED can generate more detailed and comprehensive captions, effectively capturing fine-grained details while maintaining high accuracy across visual attributes.

## 5.2 QUALITATIVE ANALYSIS

To further assess whether ED effectively mitigates object hallucination beyond quantitative metrics, we conduct qualitative evaluations using other decoding methods as baselines. As shown in Figure 6, both Regular and AGLA generate hallucinations (in red text). Specifically, a key limitation of AGLA is that it computes attention from the initial image and question only once, creating a fixed masked image by masking low-attention areas. Since this masking is static and does not dynamically update as new tokens are generated, AGLA struggles to describe fine details. In contrast, ED generates a different attention map at each token generation step and performs the logit ensemble process continuously. It allows the model to adapt to the context and better identify relevant image regions, significantly reducing hallucinations. Figure 6 illustrates ED's ability to generate dynamic attention maps at each step (in blue text). Additional qualitative examples are provided in the Appendix E.

## 5.3 COMPUTATIONAL EFFICIENCY

To evaluate the computational efficiency and performance of ED and FastED, we conduct experiments on the CHAIR benchmark, comparing them with Regular and AGLA, as shown in Table 3. AGLA is included as it is the best-performing existing method. While Regular offers the shortest inference time due to its lack of additional techniques,

Table 3: Inference latency and performance results. Latency refers to the average time per image for caption generation.

| Decoding | Latency↓ | CHAIR$s$↓ | CHAIR$l$↓ | Recall↑ |
|---|---|---|---|---|
| Regular | **5.25** | 51.0 | 15.2 | 75.2 |
| AGLA | 7.33 | **43.0** | 14.1 | 78.9 |
| FastED | 6.96 | 43.7 | **13.1** | 74.0 |
| ED | 16.42 | **43.0** | 14.0 | **82.5** |

it underperforms in object hallucination tasks. In contrast, ED achieves the highest performance (82.5% in recall) but requires the longest inference time due to multiple forward passes. FastED addresses this by reducing inference time by more than half, making it 2.36 times faster than ED and slightly faster than AGLA. FastED not only accelerates the process but also achieves significantly better CHAIR metric scores than Regular, striking a balance between speed and performance. Although AGLA also mitigates object hallucination by removing unnecessary parts and assembling logits, it relies on external modules, reducing efficiency and lacking dynamic adaptation, which leads to lower recall. Thus, ED is recommended when maximum performance and detailed results are crucial, while FastED is better suited for scenarios where inference speed is a priority.

## 5.4 ABLATION STUDIES

**Attention-Guided Weight**  To further validate the effectiveness of attention-guided weight, we assess the model's performance by assigning uniform weights to all sub-images instead of utilizing attention-guided weights. As shown in Table 4, excluding attention-guided weight results in a significant performance drop. These findings underscore the critical role of attention-guided weight in enhancing the effectiveness of ensemble decoding.

Table 4: Results on the POPE MSCOCO benchmark comparing uniform weight and attention-guided weight.

| Setting | Weighting Strategy | F1 Score | Accuracy |
|---|---|---|---|
| *Random* | Uniform | 80.68 | 83.60 |
| | Attention-Guided | **88.84** | **89.63** |
| *Popular* | Uniform | 80.02 | 82.90 |
| | Attention-Guided | **87.34** | **88.03** |
| *Adversarial* | Uniform | 78.44 | 81.23 |
| | Attention-Guided | **84.59** | **84.97** |

**ED Adaptive Plausibility Constraint**  We evaluate the effectiveness of ED adaptive plausibility constraint in object hallucination tasks, comparing it to adaptive plausibility constraint (Li et al., 2022). Figure 7 shows the averaged accuracy for both methods on the POPE evaluation. With $\beta = 0$, both methods produce the same results, as no constraint is applied. However, as $\beta$ increases,

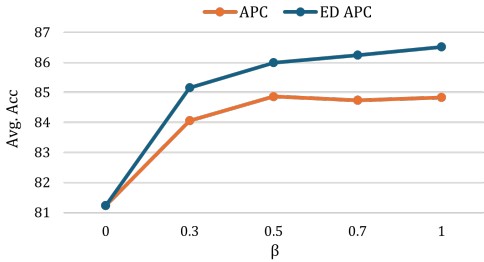
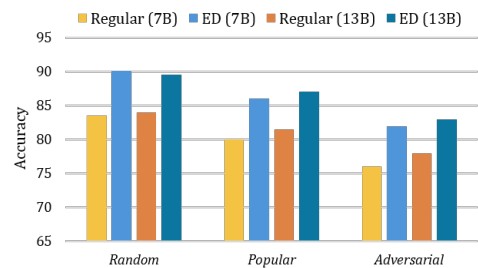

Figure 7: Ablation studies on the general adaptive plausibility constraint (APC) and ED adaptive plausibility constraint (ED APC). Averaged POPE accuracy is reported, with performance evaluated by varying $\beta$.

Figure 8: Ablation studies on model size, showing comparisons between Regular and ED for the 7B and 13B parameters across different POPE settings.

the gap between the methods widens. While adaptive plausibility constraint outperforms the no-constraint case, it consistently underperforms ED adaptive plausibility constraint at all non-zero $\beta$.

**Different Model Sizes** We perform an ablation study to evaluate the effectiveness of ED across different model sizes. Using Regular as the baseline, we measure accuracy on POPE with LLaVA-1.5 models having 7B and 13B parameters. As shown in Figure 8, ED consistently outperforms Regular, regardless of model size, confirming that our method is not limited by model capacity. Additionally, the 13B ED model generally achieves higher performance than the 7B model, indicating that ED's effectiveness improves with larger models. These results demonstrate that our approach scales well with model size while maintaining its core advantages.

# 6 RELATED WORKS

**Large Vision-Language Models** Recently, LVLMs (Liu et al., 2023b; Bai et al., 2023; Liu et al., 2024b; Dai et al., 2023; Cha et al., 2024) have emerged as pivotal innovations, combining natural language processing and computer vision to enable models to follow instructions based on visual inputs. Despite advancements in training pipelines (Chen et al., 2023b; Liu et al., 2024b) and multi-task capabilities (Chen et al., 2023a; Zhang et al., 2023b; Li et al., 2024), efficiently encoding and integrating visual information into LLMs remains one of the central challenges. LVLMs are broadly categorized by the type of projector used: the patch-wise projector (Liu et al., 2023b; 2024b; Cha et al., 2024; Chen et al., 2023a) and the resampler (Bai et al., 2023; Dai et al., 2023; Ye et al., 2023; Zhu et al., 2023). The patch-wise projector preserves spatial feature locality but incurs higher computational costs at higher resolutions due to more visual patches (Cha et al., 2024). Conversely, the resampler reduces token numbers but struggles to retain visual feature locality. We focus on the patch-wise projector method, which preserves the spatial locality of image patches, enabling effective use of attention maps.

**Object Hallucination in LVLMs** Hallucination in LLMs refers to the generation of nonsensical or nonexistent information (Ji et al., 2023; Zhou et al., 2020; Zhang et al., 2023c; Li et al., 2023a; Zhang et al., 2023a). In LVLMs, object hallucination involves incorrect descriptions of nonexistent objects or misinterpretations of image content (Biten et al., 2022; Rohrbach et al., 2018; Li et al., 2023b). Efforts to address object hallucinations have included utilizing preference optimization (Sun et al., 2023; Gunjal et al., 2024; Sarkar et al., 2024; Chen et al., 2023c; Ouali et al., 2024) and post-hoc revisers (Zhou et al., 2023; Wu et al., 2024; Yin et al., 2023), but these require supplementary data or training, which leads to increased computational costs. To tackle these issues, various decoding strategies have been proposed. VCD (Leng et al., 2024) proposes contrastive decoding that mitigates hallucinations by adding noise to images to counteract inherent model biases. Other contrastive decoding methods, such as ICD (Wang et al., 2024) and IBD (Zhu et al., 2024), have also been introduced. OPERA (Huang et al., 2024) analyzes hallucination patterns from knowledge aggregation and mitigates them by applying penalty terms to these patterns. AGLA (An et al., 2024) addresses attention deficiency with an image-prompting module that applies masks to extract local features but lacks dynamic adjustment during token generation. HALC (Chen et al., 2024) uses an

additional detector for grounding and adaptive focal contrast but is limited by its time-consuming process and dependence on external modules. In contrast, our approach enhances the LVLM's attention mechanism with a dynamic, module-free method to more effectively mitigate hallucinations.

# 7 CONCLUSION

In this paper, we propose Ensemble Decoding (ED), a novel strategy for mitigating object hallucination in LVLMs. By splitting input images into sub-images and leveraging attention maps, ED enhances logit distribution accuracy, effectively utilizing intrinsic visual information. Additionally, we introduce ED adaptive plausibility constraint to refine outputs, and FastED, a variant of ED designed for speed-critical applications. Extensive experiments across various object hallucination benchmarks demonstrate that ED achieves state-of-the-art performance, validating its effectiveness.

**Limitation** While ED significantly improves performance, it currently applies only to LVLM architectures that preserve spatial locality and use patch-wise projectors, thus limiting its applicability across all model types. Additionally, despite ED's empirical success, we lack rigorous theoretical proof explaining its effectiveness. Future work will address this, aiming to provide a stronger theoretical foundation and extend ED's applicability to more architectures.

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

# A    PILOT STUDY: OBJECT-GRID ATTENTION ALIGNMENT

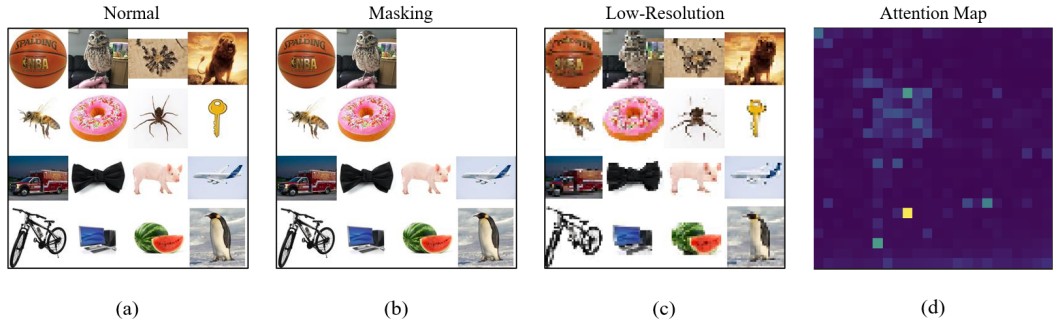

Figure 9: Example of a $4 \times 4$ grid image used in the Object-Grid Attention Alignment experiment. (a) shows the normal image, (b) illustrates the image with non-relevant areas masked, and (c) represents the image with reduced object resolutions. (d) displays the attention map from the LVLM when given the question (*Is there a donut in the image?*). The highest mean attention score corresponds to the position of the donut grid cell.

To better understand how visual characteristics influence visual representations, we conduct the Object-Grid Attention Alignment experiment, using attention maps to evaluate their impact. In this experiment, we arrange various objects in a grid format to facilitate quantitative evaluation. The grid images are created using DomainNet dataset (Peng et al., 2019), a large-scale collection of common objects across six domains. It includes *clipart, infograph, painting, quickdraw, real, and sketch* domains, with 345 object categories such as bracelets, planes, birds, and cellos. For our experiment, we specifically use the *real* domain, which consists of real-world photographs. An example of this setup is shown in Figure 9(a). We vary the number of objects and grid sizes within the images, experimenting with $2 \times 2$, $4 \times 4$, $6 \times 6$, and $8 \times 8$ grid configurations. For each grid size, we generate 100 randomly arranged images to ensure a comprehensive evaluation of how different grid sizes and object counts impact model performance. These variations allow us to assess the model's attention alignment across a range of visual complexities.

We evaluate the model's attention maps by observing the attention values corresponding to each object's position in the image. For each object, we calculate the mean attention value over its region as detailed in Equation 3. This approach allows us to evaluate the model's accuracy in correctly identifying the object based on attention alignment. Additionally, we apply two image modifications for further evaluation: (1) *Masking*, where 1/4 of the random non-relevant regions of the image were removed, and (2) *Low-Resolution*, where the resolution of objects in the image are intentionally reduced. These modifications are shown in Figure 9(b) and Figure 9(c), respectively, and they allow us to analyze the impact of visual characteristics on how LVLMs understand the representation of images in response to queries and where they focus their attention.

# B    DETAILED EXPERIMENTAL SETTINGS

Image Dimensions Distribution

Figure 10: Statistical boxplots representing the height and width distributions of MSCOCO (Lin et al., 2014) images used in the main experiment.

**The number of sub-images**    In our experiments, we set the sub-image size to $336 \times 336$ to match the resolution for which CLIP-ViT-L-336px (Radford et al., 2021), the vision encoder used in LLaVA-1.5 (Liu et al., 2024b), was pretrained and fine-tuned. Figure 10 illustrates the statistical distribution of the height and width of images used in the main experiment. The majority of images fall within the range of 400 to 600 pixels, indicating that four sub-images are sufficient for effective representation. Consequently, we established $N$ as 4 for this research.

Table 5: Results on the POPE MSCOCO benchmark with and without overlap.

| Setting | Method | F1 Score | Accuracy |
|---|---|---|---|
| *Random* | ED (w/o overlap) | 84.68 | 86.27 |
| | ED | **88.84** | **89.63** |
| *Popular* | ED (w/o overlap) | 83.47 | 84.97 |
| | ED | **87.34** | **88.03** |
| *Adversarial* | ED (w/o overlap) | 80.57 | 81.40 |
| | ED | **84.59** | **84.97** |

**Overlap between sub-images**    To ensure continuity across sub-images, we allow overlap when segmenting the original image. Specifically, for images with a width or height exceeding 672 pixels, we first resize them to $448 \times 448$ before applying our method, ensuring that adjacent sub-images share overlapping regions. This strategy helps mitigate issues where objects near the center of sub-images might otherwise be artificially split. To assess the effectiveness of this approach, we compare our method with a non-overlapping baseline, where images are simply divided into four equal parts without any overlap. As shown in Table 5, incorporating overlap prevents objects from being fragmented across sub-images and leads to a significant improvement in performance.

**Hyperparameters in FastED**    In this variant, the temperature parameter $\tau$ is not used to streamline the computation. All other parameters are kept consistent with those used in ED.

**CHAIR**    We use 100 selected images in MSCOCO (Lin et al., 2014) following the approach of An et al. (2024)*. Then, we employ the prompt *Please describe this image in detail* to generate detailed image captions. These captions are then evaluated for object hallucination. The equation used to evaluate CHAIR is presented as follows:

$$CHAIR_S = \frac{|\{\text{Captions w/ hallucinated objects}\}|}{|\{\text{All captions}\}|}, \ CHAIR_I = \frac{|\{\text{Hallucinated objects}\}|}{|\{\text{All mentioned objects}\}|} \quad (8)$$

---

*https://github.com/Lackel/AGLA

$$Recall = \frac{|\{\text{Accurate objects}\}|}{|\{\text{Ground-truth objects}\}|} \quad (9)$$

**GPT-aided Evaluation** For LLaVA-Bench (Liu et al., 2023b) GPT-aided evaluation, we use the prompt *Describe this photo in detail* to generate detailed image captions following the methodology outlined in Yin et al. (2023); Li et al. (2022); An et al. (2024). We employed the GPT-4o[†] API to assess and compare the accuracy and detailedness of the captions.

---

[†]`https://openai.com/index/gpt-4o-system-card/`

# C   ABLATION STUDIES

## C.1   WEIGHTED TERM $\alpha$ IN ED

Table 6: Results on the POPE MSCOCO *Random* with varying $\alpha$ values and without ED APC.

| $\alpha$ | F1 Score | Accuracy |
|---|---|---|
| 0 (Normal Decoding) | 79.67 | 81.40 |
| 0.3 | 80.27 | 81.80 |
| 0.5 | 80.92 | 82.22 |
| 0.7 | **81.66** | **82.73** |

We conduct additional experiments, specifically performing ablation studies in scenarios where ED adaptive plausibility constraint is not used ($\beta$=0). The results, summarized in Table 6, show that increasing $\alpha$ consistently leads to improved performance. This suggests that giving more weight to sub-images positively impacts overall model performance.

## C.2   EFFECT OF $\beta$ ON CAPTIONING ACCURACY AND DETAILEDNESS

Table 7: Results of GPT-aided evaluation on the captions from LLaVA-Bench with varying $\beta$ values.

| $\beta$ | Accuracy | Detailedness |
|---|---|---|
| 0.01 | 6.14 | 5.92 |
| 0.1 | 6.72 | **6.50** |
| 0.5 | **6.93** | 6.46 |

To further explore the optimal value of $\beta$ in ED, we conduct an experiment on image captioning using LLaVA-Bench, a task that differs in focus from POPE. In this experiment, we vary the $\beta$ while keeping other parameters constant. Along with the default setting of $\beta$=0.5, we test additional values, such as 0.1 and 0.01. As shown in Table 7, higher $\beta$ values consistently lead to improved performance in terms of accuracy. A similar trend is observed for detailedness; however, when $\beta$ is set to 0.01, the results are noticeably lower compared to $\beta$=0.1 and $\beta$=0.5, which produce nearly identical outcomes. These findings suggest that while the optimal $\beta$ value may vary across benchmarks, increasing $\beta$ generally enhances performance, contributing to more accurate and detailed captions for sub-images.

## C.3   RESULTS OF MODEL SIZE ABLATION

Table 8: Quantitative results of POPE for different model sizes.

| Setting | Model Size | Decoding Strategy | Precision | Recall | F1 Score | Accuracy |
|---|---|---|---|---|---|---|
| *Random* | 7B | Regular | 88.84 | 76.76 | 82.28 | 83.49 |
| | | **ED** | 93.40 | 86.41 | 89.68 | 90.08 |
| | 13B | Regular | 88.87 | 77.73 | 82.85 | 83.94 |
| | | **ED** | 94.05 | 84.69 | 89.05 | 89.61 |
| *Popular* | 7B | Regular | 82.47 | 76.76 | 79.34 | 79.98 |
| | | **ED** | 86.12 | 86.41 | 86.00 | 86.09 |
| | 13B | Regular | 84.30 | 77.73 | 80.77 | 81.51 |
| | | **ED** | 89.19 | 84.69 | 86.77 | 87.10 |
| *Adversarial* | 7B | Regular | 76.11 | 76.80 | 76.26 | 76.03 |
| | | **ED** | 79.75 | 86.47 | 81.90 | 82.75 |
| | 13B | Regular | 78.29 | 77.94 | 77.94 | 77.93 |
| | | **ED** | 81.95 | 85.22 | 83.34 | 82.92 |

Table 8 shows the ablation study results comparing Regular and ED across different model sizes. We evaluate both methods using LLaVA-1.5 (Liu et al., 2024b) with 7B and 13B parameters on the

POPE benchmark under random, popular, and adversarial settings. The metrics measured include precision, recall, F1 score, and accuracy for each setting. The results demonstrate that ED consistently outperforms Regular across all settings and metrics, regardless of model size. This confirms that ED is effective irrespective of model capacity, and its performance improves as the model size increases generally.

# D   MORE RESULTS ON POPE

Table 9: Detailed results of POPE using LLaVA-1.5 7B. The numbers in the table represent the mean, and the values in parentheses indicate the standard deviation.

| Dataset | Setting | Decoding Strategy | Precision | Recall | F1 Score | Accuracy |
|---------|---------|-------------------|-----------|--------|----------|----------|
| MSCOCO | Random | Regular | 92.13 (0.54) | 72.80 (0.57) | 81.33 (0.41) | 83.29 (0.35) |
| | | ED | 96.65 (0.52) | 82.00 (0.52) | **88.72** (0.18) | **89.58** (0.16) |
| | Popular | Regular | 88.93 (0.60) | 72.80 (0.57) | 80.06 (0.05) | 81.88 (0.48) |
| | | ED | 92.67 (0.18) | 82.00 (0.52) | **87.01** (0.27) | **87.76** (0.22) |
| | Adversarial | Regular | 83.06 (0.58) | 72.75 (0.59) | 77.57 (0.57) | 78.96 (0.52) |
| | | ED | 87.05 (0.76) | 82.00 (0.53) | **84.75** (0.29) | **81.90** (0.32) |
| A-OKVQA | Random | Regular | 87.24 (0.68) | 78.36 (0.54) | 82.56 (0.50) | 83.45 (0.48) |
| | | ED | 91.62 (0.42) | 89.67 (0.47) | **90.63** (0.24) | **90.73** (0.24) |
| | Popular | Regular | 80.85 (0.31) | 78.36 (0.54) | 79.59 (0.37) | 79.90 (0.33) |
| | | ED | 84.27 (0.31) | 89.67 (0.47) | **86.89** (0.10) | **86.47** (0.09) |
| | Adversarial | Regular | 72.08 (0.53) | 78.48 (0.38) | 75.15 (0.23) | 74.04 (0.34) |
| | | ED | 75.00 (0.18) | 89.78 (0.47) | **81.72** (0.31) | **79.92** (0.30) |
| GQA | Random | Regular | 87.16 (0.39) | 79.12 (0.35) | 75.15 (0.23) | 74.04 (0.34) |
| | | ED | 91.91 (0.47) | 87.58 (0.20) | **89.69** (0.33) | **89.93** (0.34) |
| | Popular | Regular | 77.64 (0.26) | 79.12 (0.35) | 78.37 (0.18) | 78.17 (0.17) |
| | | ED | 81.41 (0.75) | 87.58 (0.20) | **84.38** (0.43) | **83.79** (0.52) |
| | Adversarial | Regular | 73.19 (0.49) | 79.16 (0.35) | 76.06 (0.24) | 75.08 (0.33) |
| | | ED | 77.20 (0.30) | 87.62 (0.95) | **82.07** (0.42) | **80.87** (0.37) |

As shown in Table 9, we report the precision, recall, F1 score, and accuracy for each different setting of POPE. To verify the consistency of ED in object hallucination tasks, we repeat the implementations three times each and report the mean and standard deviation. The results demonstrate that ED consistently performs better than Regular in object hallucination tasks.

# E   PROMPT AND ADDITIONAL CASES OF GPT-AIDED EVALUATION

The prompt used for the evaluation of LLaVA-Bench[‡] captioning is presented in Table 10. It provides the instructions necessary for assessing the quality of model-generated responses. This prompt is given to GPT-4 without disclosing which model generated the captions for the two assistants, and it is scored on a scale of 1 to 10. For qualitative analysis, examples comparing Regular with ED and Regular with FastED are presented in Tables 11 and 12, respectively. From these examples, it is evident that ED and FastED produce better responses compared to Regular, as indicated by their ability to generate more accurate and detailed answers. Specifically, GPT-4 assigns significantly higher scores to ED in terms of accuracy and detailedness, validating the effectiveness of the ED approach. The same experimental settings are applied to evaluate AGLA and it enables a direct comparison across different configurations.

---

[‡]https://huggingface.co/datasets/liuhaotian/llava-bench-in-the-wild

Table 10: Prompts for GPT-aided evaluation on LLaVA-Bench.

**Instructions:**

You are an AI designed to evaluate and score the performance of two AI assistants in describing a given image. Your primary focus is on the accuracy and detailedness of their descriptions. You will assess the accuracy by checking for hallucinations—any part of the description that is inconsistent with the image content. For detailedness, you will consider how rich the response is in necessary details, excluding any hallucinated parts. You will provide scores on a scale from 1 to 10 for each assistant separately, based on these criteria. After scoring, you will offer an explanation for your evaluation, ensuring it is free from bias and not influenced by the order of presentation of the responses.

**Input format:**

```
[Assistant 1]
{Response 1}
[End of Assistant 1]
[Assistant 2]
{Response 2}
[End of Assistant 2]
```

**Output format:**

Accuracy:

Scores of the two answers:

Reason:

Detailedness:

Scores of the two answers:

Reason:

Table 11: Case of GPT-aided evaluation, where Assistant 1 is Regular and Assistant 2 is ED.

**Input:**

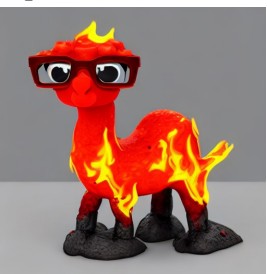

[Assistant 1]

In this image, there is a creepy sculpture or figure of a pig with bright flames or fireworks coming out of its ears and appearing to extend from its tail. The personification of the pig is quite intriguing and unique, giving it an impressive appearance. The floor and the pig sculpture both have a reflective and shiny appearance, contributing to the visual appeal of the scene. This strange creation can likely be found in a dimly-lit room or an art installation, capturing the attention and curiosity of those who encounter it.

[End of Assistant 1]

[Assistant 2]

The image features a red and orange fire-breathing lizard statue, possibly made of plastic, standing on a white surface. The lizard statue appears to be on fire, with flames coming from its mouth. It has a pair of glasses on its face, adding a unique touch to its appearance. The statue's legs are positioned in a way that it appears to be walking on the ground, giving it a dynamic and eye-catching presence.

[End of Assistant 2]

**GPT-4 Output:**

Accuracy:

Assistant 1: 2, Assistant 2: 8

Reason: Assistant 1's description contains significant inaccuracies. The sculpture is described as a pig and mentions flames or fireworks coming out of its ears and tail, which is not the case in the image. This represents a considerable amount of hallucination. On the other hand, Assistant 2 correctly identifies the main features of the image, describing it as a fire-breathing lizard statue with flames coming from various parts of its body, while also noting the glasses. These details match better with the actual image.

Detailedness:

Assistant 1: 4, Assistant 2: 6

Reason: While Assistant 1 provides a rich narrative, it is filled with inaccuracies that detract from its usefulness. The mention of a *creepy sculpture*, a *dimly-lit room or an art installation*, and *a reflective and shiny appearance* of the floor add unnecessary and incorrect details that do not match the image. Assistant 2's description, though somewhat simpler, is more accurate and focuses on pertinent aspects such as the lizard's color, material, and the presence of glasses. However, it could have included more detailed observations about the surface and environment found in the image. Therefore, Assistant 2 receives a better score for its detailedness.

Table 12: Case of GPT-aided evaluation, where Assistant 1 is Regular and Assistant 2 is FastED.

**Input:**

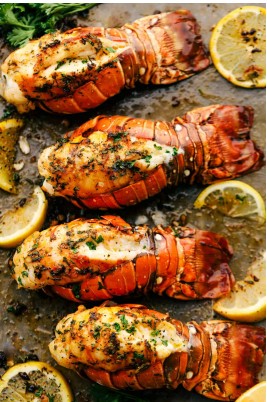

[Assistant 1]

The image features a table with a delicious meal being prepared, showcasing a variety of cooked and sliced seafood. Steamed crayfish on the table, accompanied by lemon wedges, add a lemon zest and freshness to the meal. There are five large crayfish in the image, with three of them laying in close proximity next to each other, and the remaining two placed further apart. Alternatively, they can be seen alongside five fresh lemon wedges, with some placed directly on top of the crayfish. The combination of the seafood and the lemon enhances the overall appeal and taste of this seafood meal.

[End of Assistant 1]

[Assistant 2]

The image features a close-up of several pieces of cooked lobster, likely grilled or roasted, with a generous amount of seasoning. The lobster pieces are placed on a tray, and there are a couple of lemon wedges scattered around the tray, adding a touch of freshness and acidity to the dish. The lobster pieces are arranged in various positions, with some overlapping and others sitting next to each other. The overall presentation of the lobster and lemon wedges creates an appetizing and inviting scene.

[End of Assistant 2]

**GPT-4 Output:**

Accuracy:

Assistant 1: 6, Assistant 2: 9

Reason: Assistant 1 incorrectly identifies the seafood as crayfish instead of lobster. Additionally, there is a mention of five crayfish, but the image clearly depicts only four lobster tails. Assistant 2 correctly identifies the seafood as lobster and describes the seasoning, which is consistent with the image, though the specifics of the cooking method (grilled or roasted) are not directly clear.

Detailedness:

Assistant 1: 7, Assistant 2: 8

Reason: Assistant 1 provides information about the arrangement of the seafood and the presence of lemon wedges, albeit with some inaccuracies regarding the number of items and the type of seafood. Assistant 2 gives a more accurate and detailed description of the lobster's appearance and seasoning, though lacks some detail about the exact arrangement or the garnishes present, compared to Assistant 1. Both assistants mention the lemon wedges and the general presentation, but Assistant 2's description aligns more closely with the actual content of the image.

