# OpenReview forum: "Do You Keep an Eye on What I Ask? Mitigating Multimodal Hallucination via Attention-Guided Ensemble Decoding"
_ICLR.cc/2025/Conference — ICLR 2025 Poster_

### Official Review · Reviewer_UHSv · 2024-10-22

**Soundness:** 3
**Presentation:** 4
**Contribution:** 2
**Rating:** 6
**Confidence:** 4

**Summary:**

This paper introduces a technique (ED) to reduce hallucination in multi-modal large language models (MLLMs) by splitting an image into N sub-images followed by using an attention guided ensemble decoding approach which ensembles the logits for the next token to be predicted by the MLLM using the attention weights of the original full-sized image and the N sub-images with the text prior. The authors conduct a systematic study which shows the two main causes of object hallucination in MLLMs as the number of objects in an image and the image resolution. They also provide systematic quantitative metrics which show improvement on hallucination benchmarks compared to existing SoTA methods and regular decoding approaches. The authors present a latency analysis of their method compared to other decoding based methods and provide a alternative baseline of FastED which balances accuracy with speed. Overall the paper is well written and conducts systematic experimental analysis.

**Strengths:**

1) The paper is very well written and it is easy to understand their method.
2) The approach presented is easy to implement, is training-free and only requires inference time re-weighting of logits.
2) The pilot study which talks about the main causes of object hallucination in MLLMs is instructive and useful to the community as a whole.
3) It is easy to understand the motivation of their method from the conclusions of the study they present, which shows that masking irrelevant regions of an image helps reduce object hallucination, and their approach of ensembling logits from various sub-images should help with that
4) The results achieve SoTA performance on commonly used benchmarks to measure hallucination (POPE and CHAIR)

**Weaknesses:**

1) While the idea of splitting the original image into N sub-images is well motivated from the pilot study, the approach to utilize the attention weights to re-weight the logits lacks theoretical/empirical justification. Even intuitively, it would make sense to use attention weights for prompts such as "is there a spoon in the image" to put a higher emphasis on the sub-image which contains a spoon. However, for generic prompts such as "describe the image", the attention weights across the sub-images should be more or less equal. In such cases I am not sure of the reason why it might be useful to weight the logits by the attention scores.
2) Adding on to 1, the paper lacks experiments/ablations which justify the use of attention weights for the decoding process. Adding ablations which vary the temperature parameter might be useful to understand the impact of attention on results. For example, a useful result to conclude the validity of the attention weights could be setting very high values of temperature parameter leading to sampling from uniform distribution across the sub-images.
3) The paper does not include experiments on tasks other than hallucination and image captioning. For example on the MME benchmark, out of the 16 tasks, the authors only include results on the existence, count, position, and color sub-categories. It would be good to look at the results on some of the other sub-tasks such as code reasoning, OCR, common-sense reasoning, text translation etc. (as done in the VCD paper) to ensure that this method of decoding does not regress the performance on such benchmarks.
4) The authors show results for FastED only on CHAIR and LLava-Bench, it would be good to see results of FastED on POPE and MME benchmarks as well
5) This is already mentioned in the limitations section, but the method only works for MLLM architectures which use a linear projector and does not work with resampler and Q-former based architectures (BLIP-2, Qwen-VL). This limits the generalizability of the technique across a wide-range of model architectures.

**Questions:**

Please refer to the weaknesses section for questions. Additionally:

1) Have you tried the recent Qwen-2 VL model with your technique? Given that it does not use a Q-former based technique, it would be interesting to see how this method works with a model architecture other than LLava
2) If possible, could you provide some additional examples of attention maps and final attention weights for the N sub-images for different sets of prompts and input images? It would be helpful to understand how exactly the attention weights vary.

---

> ### Author Response · Authors · 2024-11-17
> **Response to Reviewer UHSv (1/2)**
>
> Thank you for your positive evaluation of our pilot study and review of our method. Your detailed feedback has greatly contributed to deepening our research. We’ve prepared responses to your feedback and would appreciate it if you could take a look.
>
> ---
>
> ## **Weaknesses**
>
> ### **4.1 Impact of Prior Tokens on Attention-Guided Weights**
>
> Our attention-guided weights are influenced not only by the prompt but also by the preceding output tokens. This means that all prior tokens are considered when calculating the attention-guided weights and generating the next token. The examples in Figure 6 demonstrate how attention weights are dynamically allocated to the most relevant regions just before the next token is generated. While the prompt remains fixed, the attention map continuously adapts based on the focus of the preceding tokens. This characteristic allows the model to produce more accurate and detailed descriptions in captioning tasks, as demonstrated by the improved performance on the CHAIR and LLaVA-bench captioning benchmarks.
>
> ----
>
> ### **4.2 Validation of Attention-Guided Weights**
> To further validate the effectiveness of attention-guided weights, we evaluate the model's performance by assigning equal weights to all sub-images instead of using attention-guided weights. The results in Table R1 reveal a significant drop in performance when attention-guided weights are excluded. Additionally, we examine how varying the temperature parameter ($\tau$) impacts performance, observing a trend where lower temperatures generally improve outcomes. However, no substantial visual differences are detected in the outputs.
>
> **Table R1. Results on the POPE COCO benchmark with and without attention-guided weights.**
> | Setting       | Method                        | F1 Score | Accuracy |
> |---------------|-------------------------------|----------|----------|
> | **Random**    | ED (w/o attention-guided weight) | 80.68    | 83.6     |
> |               | ED ($\tau$ = 1e-1)                           | 88.27    | 89.23    |
> |               | ED ($\tau$ = 1e-2)                             | **88.84**    | **89.63**    |
> | **Popular**   | ED (w/o attention-guided weight) | 80.02    | 82.9     |
> |               | ED ($\tau$ = 1e-1)                             | 86.91    | 87.80   |
> |               | ED ($\tau$ = 1e-2)                             | **87.34**    | **88.03**   |
> | **Adversarial** | ED (w/o attention-guided weight) | 78.44    | 81.23    |
> |               | ED ($\tau$ = 1e-1)                             | 84.53    | **85.17**    |
> |               | ED ($\tau$ = 1e-2)                             | **84.59**    | 84.97    |
>
> ----
>
> ### **4.3 ED's Applicability to Other Tasks**
> To further evaluate performance beyond hallucination and image captioning, we conduct experiments on other tasks in the MME benchmark. As described in Table R2, ED shows a noticeable improvement in tasks such as OCR, achieving a higher total score compared to the previous work, VCD. This result demonstrates that our approach is not only effective in addressing hallucination and image captioning but also transferable to other tasks.
>
> **Table R2. Results on the MME benchmark comparing VCD and ED, evaluated with Accuracy+.**
> | Task      | VCD           | ED |
> |---------------|-----------|----------|
> |Poster|132.11|**136.06**|
> |Celebrity|**120.94**|119.12|
> |Scene|152.20|**153.67**|
> |Landmark|140.45|**149.67**|
> |Artwork|109.60|**116.83**|
> |OCR|104.00|**125.83**|
> |Common Sense Reasoning|111.29|**116.90**|
> |Numverical Calculation|**68.50**|52.83|
> |Text Translation|**89.50**|85.83|
> |Code Reasoning|84.00|**91.67**|
> |Total|1112.59|**1148.41**|
>
> ----
>
> ### **4.4 Performance of FastED**
> Equation R1. $f_{k,t} = \frac{\exp(s_{k,t}/\tau)}{\sum_{j=1}^N \exp(s_{j,t}/\tau)}$
>
> As shown in Equation R1, decreasing the temperature value ($\tau$) enhances the focus on the sub-image with the highest attention score among the $n$ sub-images. This adjustment causes ED to produce results that closely align with those of FastED. Additionally, for tasks requiring concise answers, such as POPE or MME, the difference in inference latency between ED and FastED is negligible, making ED a suitable choice for such tasks. As a result, we plan to use results from ED in place of FastED for these scenarios.
>
> ----
>
> ### **4.5 Expanding Applicability to Diverse VLM Architectures**
> Recent studies have highlighted the emergence of new VLMs that utilize linear projectors, offering opportunities to extend our method to a wider range of models [1-2]. Furthermore, VLMs employing resamplers (e.g., InstructBLIP, Qwen-VL) could adopt a similar approach by integrating an additional module to compute attention weights. While existing research on attention map formation is well-suited for prompts [3], there has been limited investigation into the use of dynamic attention maps at the token level within VLMs. We plan to address this gap as part of our future work.

---

> ### Author Response · Authors · 2024-11-17
> **Response to Reviewer UHSv (2/2)**
>
> ## **Questions**
> ### **4.1 Applying ED to Qwen-VL2**
>
> We attempted to apply ED to Qwen-2 VL. However, although it has the same number of parameters as LLaVA-1.5 (7B), it demands significantly more GPU resources, particularly VRAM, due to its use of Naive Dynamic Resolution. Unfortunately, our current GPU resources are insufficient to run experiments on this model. Therefore, we plan to explore extending the method to Qwen-2 VL and similar models as part of future work once sufficient resources are available.
>
> ----
> ### **4.2 Additional Examples of Attention Maps**
> To provide a clearer understanding, we have prepared an additional example demonstrating how attention scores and attention-guided weights function and the specific values they take. Please refer to the image provided via the link below. The following results illustrate the inference generated using ED with the given prompt.
>
> It is important to note that the image linked below is not associated with the authors and was randomly selected from the LLaVA-Bench benchmark dataset.
>
> - Image link: https://huggingface.co/datasets/liuhaotian/llava-bench-in-the-wild/blob/main/images/024.jpg
>
> - Prompt: Describe this photo in detail.
>
> - ED Result: The image depicts a busy highway at night, filled with numerous cars driving in both directions. The traffic is quite dense, with vehicles occupying almost every lane of the highway. The cars are spread across the entire width of the highway, with some cars closer to the ***foreground*** and others further back, creating a sense of depth in the scene. The highway appears to be a part of a city, as there are several ***pedestrians*** walking along the sidewalk, some closer to the foreground and others further back. The overall atmosphere of the image is bustling and energetic, reflecting the typical nighttime traffic in a ***large*** city.
>
>
> As shown in Figure 6, the attention map dynamically adjusts based on the prompt and preceding tokens. To illustrate this, we have summarized the specific attention scores and attention-guided weights prior to generating three key tokens in Table R3 and Table R4, respectively.
>
> In Table R4, the attention-guided weights reveal distinct patterns. Before generating the token 'foreground', the weights are concentrated in the bottom regions (Left-bottom and Right-bottom), with particularly high values on the left side where more cars are located. For the token 'pedestrian', which corresponds to the right-bottom area of the image, the weights are sharply focused on that region. Finally, when generating tokens like 'large', which require a broader perspective of the entire image, the weights are distributed more evenly across all regions. In summary, the attention-guided weights in ED effectively highlight the importance of the relevant sub-images at each stage of generation. We hope this example provides clarity and enhances your understanding of the mechanism.
>
>
>
> **Table R3. Attention scores for four regions before the generation of each word token.**
> | Token| Left-top | Right-top | Left-bottom | Right-bottom |
> |-----------|----------|------------|-------------|--------------|
> | foreground | 0.18 | 0.18 | **0.24** | 0.22 |
> | pedestrians | 0.14 | 0.17 | 0.19 | **0.81** |
> | large | 0.16 | **0.20** | 0.18 | 0.19 |
>
> **Table R4. Attention-guided weights for four regions before the generation of each word token.**
> | Token| Left-top | Right-top | Left-bottom | Right-bottom |
> |-----------|----------|------------|-------------|--------------|
> | foreground | 0.00 | 0.00 | **0.85** | 0.14 |
> | pedestrians | 0.00 | 0.00 | 0.00 | **1.00** |
> | large | 0.01 | **0.52** | 0.13 | 0.34 |
>
>
> ----
>
>
> [1] Li, B., Zhang, Y., Guo, D., Zhang, R., Li, F., Zhang, H., ... & Li, C. (2024). Llava-onevision: Easy visual task transfer. arXiv preprint arXiv:2408.03326.
>
> [2] Wang, P., Bai, S., Tan, S., Wang, S., Fan, Z., Bai, J., ... & Lin, J. (2024). Qwen2-vl: Enhancing vision-language model's perception of the world at any resolution. arXiv preprint arXiv:2409.12191.
>
> [3] An, W., Tian, F., Leng, S., Nie, J., Lin, H., Wang, Q., ... & Lu, S. (2024). AGLA: Mitigating Object Hallucinations in Large Vision-Language Models with Assembly of Global and Local Attention. arXiv preprint arXiv:2406.12718.

---

> > ### Comment · Reviewer_UHSv · 2024-11-23
> >
> > Thanks for addressing the questions I had. Looking at Table R1 and Table R2 it does seem like using attention weights compared to using uniform weights and also that using ED leads to a general increase in performance compared to VCD. The attention values for the 4 sub-images also show that there is relevant focus on the corresponding sub-image while generation, which aligns well with the claims made in the pilot-study. I will increase my rating to 6 as the method shows promising results.
> >
> > Follow up questions:
> >
> > 1) Is there a reason for the performance drop in Numerical calculation and text translation? Is it because the splitting leads to issues with parsing text/equation heavy images?

---

> ### Author Response · Authors · 2024-11-21
>
> ### **4.4 Performance of FastED**
> Thanks to your valuable feedback, we conducted experiments on FastED using hallucination benchmark datasets, POPE and MME, and would like to report the findings. The results are summarized in Tables R5 and R6.
> In the POPE dataset, FastED performs nearly on par with ED. However, in the MME dataset, its performance is slightly lower than that of ED. While FastED works well for tasks requiring concise answers, ED proves to be more effective overall when tasks demanding longer responses are considered. Nevertheless, for tasks that require short answers and prioritize faster inference time, FastED serves as a practical alternative.
>
> **Table R5. Results of ED and FastED on the POPE COCO benchmark.**
> |Setting | Method | F1 Score | Accuracy |
> |--------|--------|----------|----------|
> | Random| ED | 88.84 | 89.63 |
> || FastED | 88.99 | 89.77 |
> |Popular| ED | 87.34 | 88.03 |
> || FastED | 87.43 | 88.10 |
> |Adversarial| ED | 84.59 | 84.97 |
> ||FastED | 84.71 | 85.07 |
>
> **Table R6. Results of ED and FastED on the MME Hallucination benchmark.**
> | Method | Existence | Count | Position | Color | Total |
> |--------|------------|-------|----------|--------|-------|
> | ED | 190.00 | 129.47 | 135.00 | 164.97 | 619.43 |
> | FastED | 188.33 | 137.78 | 126.11 | 163.33 | 615.55 |

---

> ### Author Response · Authors · 2024-11-24
>
> Thank you so much for your positive evaluation. Your thoughtful feedback has been incredibly valuable in improving our research.
>
> ED is a method that decodes by ensemble processing split sub-images. However, splitting images containing entire characters or numbers that must remain intact can disrupt the original semantic meaning, similar to fragmenting a string. This limitation likely contributed to poorer performance in tasks where such integrity is crucial. To improve performance across the aforementioned tasks, a potential future research direction could involve dynamically adjusting sub-image sizes to ensure that text or characters are not fragmented.

---

### Official Review · Reviewer_98vf · 2024-10-31

**Soundness:** 3
**Presentation:** 3
**Contribution:** 2
**Rating:** 6
**Confidence:** 3

**Summary:**

Large Vision-Language Models (LVLMs) excel at tasks like image captioning but face challenges with object hallucination, where they describe objects that aren't actually present in images. While existing solutions like data augmentation have attempted to address this issue, they face scalability problems and often require additional modules. This paper introduces Ensemble Decoding (ED), which works by dividing input images into smaller parts and combining their logit distributions using attention map-based weighting. It also develops ED adaptive plausibility constraint for logit distribution calibration and a faster variant called FastED for time-sensitive applications. Through extensive testing on hallucination benchmarks, the proposed method demonstrates superior performance compared to existing approaches.

**Strengths:**

1. It is interesting to explore the effect of the number of unnecessary objects in an image and the object resolution on the performance of LVLM.
2. The proposed method is simple and straightforward to implement, which enhances its reproducibility.
3. Experiments on multiple benchmarks demonstrate that the proposed method can achieve better performance than state-of-the-art work.

**Weaknesses:**

I have some concerns about this paper:

1. From Figure 3, it seems that feeding multiple sub-images into LVLM gives more correct answers than feeding the original image. I'm curious about the performance of ED without using the original image.
2. If LVLM produces the correct output for the original image, but produces the wrong output for the sub-image, does ED negatively affect the understanding of the model in this case? For example, can the split sub-image get a valid output when the target object is located at the centre of the image?
3. It is interesting that low-resolution objects may trigger the hallucination of large models. However, dividing the image into small sub-images does not improve the absolute resolution of the object but rather the percentage of the object in the image.
4. How is the metric of inference latency calculated? ED only slightly outperforms AGLA in terms of recall at the cost of more than twice the inference latency; FastED is close to AGLA in terms of efficiency but has lower recall. So what is the advantage of the method proposed in this paper over AGLA? The authors only show the performance of FastED on the CHAIR dataset, which makes its performance difficult to evaluate.
5. In this paper, there is a lack of analysis and ablation experiments on the impact of the constituent modules in ED (e.g. Attention-Guided Weight, and Adaptive Plausibility Constraint) on the performance of the method.
6.The exact number of sub-images into which the image is split (determined by the hyperparameter N) may have a strong impact on the efficiency and performance of the ED, and the impact of this factor is not analysed in this paper.
7. From Table 4, the improvement from aggregating the logits of sub-images does not seem to be significant.

**Questions:**

When splitting the original image into sub-images, is there no overlap at all between each sub-image?

---

> ### Author Response · Authors · 2024-11-17
> **Response to Reviewer 98vf (1/2)**
>
> Thank you for your insightful review, which helped refine and strengthen our work. We hope our detailed responses address your concerns.
>
> ---
>
> ## **Weaknesses**
>
> ### **3.1 Performance of ED without Original Image**
>
> As explained in Equation 6, when $\beta = 1$ in our proposed ED Adaptive Plausibility Constraint (APC), only the outputs of the multiple sub-images are utilized, completely excluding the outputs of the original image. The performance of ED APC with $\beta = 1$ is shown in Figure 7. Under this setting, the average accuracy in POPE is the highest among all configurations, demonstrating that the outputs from multiple sub-images are sufficient to produce more accurate answers.
>
> In our study, we set the $\beta$ to 0.5 to enhance generalization across various tasks, including scenarios where objects are incorrectly split or the overall structure of the image is critical, while balancing performance across multiple benchmarks. This effectively enables the utilization of outputs from both the sub-images and the original image.
>
> ---
>
> ### **3.2 Negative Effects of Sub-Image Outputs**
>
> Such cases are possible, and we carefully consider this issue. To address it, as shown in Appendix B, we intentionally adjust the process to allow overlap between sub-images to prevent objects from being cut off (for more details on overlap between sub-images, please refer to **Questions 3.1**).
>
> As shown in Figure 7, the results illustrate that as the $\beta$ increases, signifying a greater reliance on sub-images compared to the original image, POPE performance improves. This suggests that sub-images are more likely to contribute beneficial information rather than introduce negative effects. Furthermore, we conduct additional experiments varying $\alpha$ when ED APC is not applied ($\beta=0$). As shown in Table R1, increasing $\alpha$ consistently led to significant performance improvements. This finding reinforces the conclusion that emphasizing sub-images has a positive effect on overall model performance.
>
> **Table R1. Results on the POPE COCO benchmark with varying alpha values and without ED APC.**
> | α | F1 Score | Accuracy |
> |--------------------|----------|----------|
> | 0 (Normal Decoding)| 79.67 | 81.40 |
> | 0.3 | 80.27 | 81.80 |
> | 0.5 | 80.92 | 82.22 |
> | 0.7 | **81.66** | **82.73** |
>
> ---
>
> ### **3.3 Maintaining Object Resolution through Image Splitting**
>
> Our goal in splitting the image is to prioritize the resolution of specific objects that are important or relevant to the question, rather than focusing on the entire image. In our experiments, most images ranged in size from 400 to 600 pixels in width and height. Since the image encoder’s input size is fixed at $336 \times 336$, the conventional approach of resizing the entire image to fit this size significantly reduces the resolution of specific objects. To address this, we designed a splitting approach that preserves the resolution of target objects, allowing the model to process them with greater clarity.
>
> ---
>
> ### **3.4 Efficiency and Accuracy of ED and FastED**
>
> Inference latency was measured as the average time to generate a single image caption under the same conditions. When examining the performance of ED and FastED more closely, both showed strong results in GPT-aided evaluation. Figure 5 presents the results of GPT's accuracy and detailedness measurements for captioning on LLaVA-Bench. ED and FastED outperformed AGLA, with ED showing particularly significant improvements in detailedness.
>
> Additionally, in tasks requiring short answers, such as POPE and MME, ED demonstrated superior performance compared to previous methods. This suggests that our approach offers a more comprehensive improvement over existing methods. Lastly, unlike AGLA, ED and FastED do not rely on external modules. This can offer advantages such as reduced latency, simplified design, and improved scalability.

---

> ### Author Response · Authors · 2024-11-17
> **Response to Reviewer 98vf (2/2)**
>
> ### **3.5 Ablation Studies about Adaptive Plausibility Constraint and Attention-Guided Weights**
>
> The results of ablation studies about Adaptive Plausibility Constraint (APC) is presented in Figure 7. The figure reports the performance differences between the general APC proposed in prior research [1] and ED APC proposed in this study across varying $\beta$ values. On the x-axis of the figure, the $\beta$ values are shown, where $\beta=0$ represents the case where no constraint is applied during the decoding process. The results show that applying ED APC leads to better performance compared to using no constraint or the general APC, demonstrating the effectiveness of ED APC.
>
> We also conducted an additional ablation study to evaluate the effectiveness of attention-guided weights. We assigned equal weights to all sub-images and compared the results with those obtained when attention-guided weights were used. As shown in Table R2, a significant drop in performance is observed when attention weights are not applied, emphasizing their crucial role in enhancing performance.
>
> **Table R2. Results on the POPE COCO benchmark with and without attention-guided weights.**
>
> | Setting        | Method                          | F1 Score | Accuracy |
> |----------------|---------------------------------|----------|----------|
> | **Random**     | ED (w/o attention-guided weight) | 80.68    | 83.6     |
> |                | ED                               | **88.84**| **89.63**|
> | **Popular**    | ED (w/o attention-guided weight) | 80.02    | 82.9     |
> |                | ED                               | **87.34**| **88.03**|
> | **Adversarial**| ED (w/o attention-guided weight) | 78.44    | 81.23    |
> |                | ED                               | **84.59**| **84.97**|
>
> ---
>
> ### **3.6 The Number of Sub-Images**
>
> As described in Appendix B, we selected the number of sub-images used in the main experiments to be 4, considering the size of most images (400–600 in width and height) and the vision encoder's input size ($336 \times 336$). Increasing the number of sub-images can help focus on smaller objects, but it also significantly increases computational costs. Additionally, it may result in each sub-image becoming too small, requiring upscaling when inputting the cropped images into the model, which could degrade performance. While this approach may be effective for datasets with larger image sizes, we leave this exploration for future work.
>
> ---
>
> ### **3.7 Improvement from Aggregating the Logits of Sub-Images**
>
> In Table 4, we observe that, due to the influence of ED APC, the results remain robust regardless of the $\alpha$. To further investigate the impact of it, we conduct an additional experiment by removing APC and varying the value of $\alpha$. Table R1 indicates that performance improves significantly while increasing $\alpha$ without APC. Also, as $\alpha$ increases and the emphasis on sub-image outputs grows, performance improves. This highlights the importance of sub-image aggregation under these conditions.
>
> **Table R1. Results on the POPE COCO benchmark with varying $\alpha$ values and without ED APC.**
>
> | $\alpha$     | F1 Score | Accuracy |
> |-----------------|----------|----------|
> | 0 (Normal Decoding)| 79.67   | 81.40   |
> | 0.3             | 80.27    | 81.80   |
> | 0.5             | 80.92    | 82.22   |
> | 0.7             | **81.66**| **82.73**|
>
> ---
>
> ## **Questions**
>
> ### **3.1 Overlap Between Sub-Images**
>
> We deliberately allow overlap when creating the sub-images. Specifically, as outlined in Appendix B, for images with widths or heights larger than 672 pixels, we resized them to $448 \times 448$, ensuring overlap between the sub-images. This strategy was designed to mitigate issues arising from objects being split when located at the center of the sub-images.
>
> To evaluate the effectiveness of overlap between sub-images, we compare our approach with a method where the image is simply divided into four parts without overlap. As shown in Table R4, allowing overlap prevents objects from being improperly split and significantly improves performance.
>
> **Table R4. Results on the POPE COCO benchmark with and without overlap.**
>
> | Setting        | Method                    | F1 Score | Accuracy |
> |----------------|---------------------------|----------|----------|
> | **Random**     | ED (w/o overlap)          | 84.68    | 86.27    |
> |                | ED                         | **88.84**| **89.63**|
> | **Popular**    | ED (w/o overlap)          | 83.47    | 84.97    |
> |                | ED                         | **87.34**| **88.03**|
> | **Adversarial**| ED (w/o overlap)          | 80.57    | 81.40    |
> |                | ED                         | **84.59**| **84.97**|
>
> ---
>
>
> [1] Li, X. L., Holtzman, A., Fried, D., Liang, P., Eisner, J., Hashimoto, T., ... & Lewis, M. (2022). Contrastive decoding: Open-ended text generation as optimization. *arXiv preprint arXiv:2210.15097*.

---

> > ### Comment · Reviewer_98vf · 2024-11-19
> > **Response to Rebuttal**
> >
> > Thanks for the authors reply, which addresses some of my concerns. I have the following questions about this response:
> >
> > - Regarding weakness 3.1, according to the author's response, is the optimal value of $\beta$ not consistent across benchmarks? Does the performance of ED on other evaluation benchmarks show a trend of increasing with increasing $\beta$?
> > - My confusion about weakness 3.3 is that the solution of ED is not consistent with the analysis of resolution changes in the pilot study. A clearer clarification of the insights that the analysis of changes in object resolution in the pilot study brings to the design of the method may help to resolve this issue.
> > - From the available experiments, ED only shows a significant performance advantage on some benchmarks while increasing inference latency.FastED may have the potential to perform well on both performance and inference latency. However, the authors do not demonstrate FastED's performance on most experimental benchmarks or scenarios.
> >
> > I tend to keep my rating for the above reasons.

---

> ### Author Response · Authors · 2024-11-21
>
> We sincerely appreciate you taking the time to review our response carefully and for asking such insightful follow-up questions. We have prepared responses to the questions below and truly hope they resolve any remaining uncertainties you may have.
>
> ---
>
> ### **3.8 Optimal Value of beta**
>
> To further explore the optimal value of $\beta$, we conduct an additional experiment on image captioning using LLaVA-Bench, a task with a different focus from POPE. We vary the $\beta$ values while keeping other parameter settings unchanged. Alongside the default setting of $\beta$=0.5, we test additional values such as 0.1 and 0.01.
> As shown in Table R5, higher $\beta$ values consistently leed to improved performance in terms of accuracy. A similar trend is observed for detailedness, though when beta is set to 0.01, the results are noticeably lower compared to 0.1 and 0.5, which produce almost identical outcomes. These findings suggest that while the optimal beta value may vary across benchmarks, increasing $\beta$ tends to enhance performance, helping generate more accurate and detailed captions for sub-images.
>
> **Table R5. Results of GPT-aided evaluation on the captions from LLaVA-Bench with varying beta values.**
>
> | Method | Accuracy | Detailedness |
> |--------|----------|--------------|
> | ED ($\beta$ = 0.01) | 6.14 | 5.92 |
> | ED ($\beta$ = 0.1) | 6.72 | 6.50 |
> | ED ($\beta$ = 0.5) | 6.93 | 6.46 |
>
> ---
> ### **3.9 High Resolution Achieved via ED**
> In our pilot study, we intentionally reduced the resolution of individual objects within images to investigate the impact of resolution on object hallucination. Through this experiment, we observed that low object resolution exacerbates object hallucination, and this effect becomes more pronounced as the number of objects in the image increases. In most VLMs (as demonstrated in this experiment with LLaVA 1.5), the input image size for the vision encoder is fixed at 336 × 336, requiring the majority of images to be resized to smaller dimensions. This motivates us to explore methods for mitigating the resolution loss caused by this resizing process under such constraints.
>
> Our proposed approach involves cropping the original image to create sub-images. By doing so, objects either relevant to the question or identified as significant by the model are preserved within the cropped sub-image. These cropped images already match the encoder's input size, eliminating the need for the typical resizing process that reduces the original image. Consequently, unlike using the original image as input, this approach ensures that the resolution of relevant objects is preserved, maintaining the original quality of the source image.
>
>
> Thank you for pointing out this important aspect. We will ensure that this section is clarified and revised more thoroughly in the camera-ready version.
>
> ---
>
> ### **3.10 Performance of FastED**
>
> To gain a deeper understanding of FastED, we conduct additional experiments on datasets requiring concise answers, such as POPE and the MME hallucination dataset. The results are presented in Tables R6 and R7.
> Surprisingly, FastED performs nearly on par with ED on the POPE dataset. Conversely, on the MME dataset, FastED shows slightly lower performance compared to ED. These results indicate that, compared to ED, FastED tends to perform less effectively in tasks requiring longer answers, such as image captioning, while demonstrating comparable or slightly lower performance in tasks demanding shorter answers or prioritizing time efficiency. Based on these findings, it is evident that FastED is an effective model, offering both adequate performance and fast inference speed.
>
> **Table R6. Results of ED and FastED on the POPE COCO benchmark.**
> |Setting | Method | F1 Score | Accuracy |
> |--------|--------|----------|----------|
> | Random| ED | 88.84 | 89.63 |
> || FastED | 88.99 | 89.77 |
> |Popular| ED | 87.34 | 88.03 |
> || FastED | 87.43 | 88.10 |
> |Adversarial| ED | 84.59 | 84.97 |
> ||FastED | 84.71 | 85.07 |
>
> **Table R7. Results of ED and FastED on the MME Hallucination benchmark.**
> | Method | Existence | Count | Position | Color | Total |
> |--------|------------|-------|----------|--------|-------|
> | ED | 190.00 | 129.47 | 135.00 | 164.97 | 619.43 |
> | FastED | 188.33 | 137.78 | 126.11 | 163.33 | 615.55 |
>
> ---

---

> > ### Comment · Reviewer_98vf · 2024-11-22
> > **Official Comment by Reviewer**
> >
> > Thanks for your response.
> >
> > Regarding 3.8, I'm curious if the continually increasing $\beta$ on LLaVa-Bench will further improve ED's performance?

---

> ### Author Response · Authors · 2024-11-23
>
> Thank you for your dedicated and thoughtful engagement in the review process.
>
> ### **3.8 Optimal Value of beta**
> We conduct additional experiments by increasing the $\beta$ value to 0.7 and the maximum value of 1.0. As reported in Table R8, the results show that when $\beta$ is small (0.01), both accuracy and detailedness record the lowest scores. Generally, as $\beta$ increases, both metrics improve; however, beyond a certain point, the improvements are limited. When beta reaches 1.0, both accuracy and detailedness slightly decline. This suggests that for captioning tasks requiring longer responses, while the outputs of sub-images provide more accurate and detailed information, the output related to the original image also plays an essential role. Therefore, the logit ensemble process in ED must necessarily account for the outputs derived from the original image.
>
> **Table R8. Results of GPT-aided evaluation on the captions from LLaVA-Bench with varying $\beta$ values.**
> | Method | Accuracy | Detailedness |
> |--------|----------|--------------|
> | ED ($\beta$= 0.01) | 6.14 | 5.92 |
> | ED ($\beta$= 0.1) | 6.72 | 6.50 |
> | ED ($\beta$= 0.5) | 6.93 | 6.46 |
> | ED ($\beta$= 0.7) | 6.83 | 6.57 |
> | ED ($\beta$= 1.0) | 6.74 | 6.30 |

---

> > ### Comment · Reviewer_98vf · 2024-11-24
> > **Official Comment by Reviewer**
> >
> > Thanks for your response. My concerns have been largely addressed. I will raise my rating to 6.

---

> > > ### Author Response · Authors · 2024-11-24
> > >
> > > Thank you for positively reassessing our work. Your genuine, active, and insightful review has provided us with valuable motivation to further develop and improve our research.

---

### Official Review · Reviewer_mCVa · 2024-11-03

**Soundness:** 3
**Presentation:** 4
**Contribution:** 3
**Rating:** 6
**Confidence:** 3

**Summary:**

This paper addresses the issue of object hallucination in Large Vision-Language Models (LVLMs), where models generate descriptions that include nonexistent objects or misrepresent existing ones. While previous approaches such as data augmentation and training-free methods have attempted to mitigate this problem, they face scalability issues and often rely on external modules. The authors propose a new method called Ensemble Decoding (ED), which divides input images into sub-images and combines logit distributions by weighting them through an attention map. Additionally, the paper introduces ED’s adaptive plausibility constraint to calibrate logit distributions and a variant named FastED for speed-critical applications. Extensive experiments demonstrate that this method achieves state-of-the-art performance on hallucination benchmarks, confirming its effectiveness in reducing object hallucinations.

**Strengths:**

- The idea of Ensemble Decoding (ED) is interesting, and well motived by evidences of model will get right answer after applying crop and resize to the image and the toy experiment of checking whether properly divided sub-images can reduce object hallucination in the outputs o LVLMs
- The authors conducted extensive ablation studies and main experiments to validate the effectiveness of the method, and the experimental results appear to be quite convincing.

**Weaknesses:**

- There doesn’t seem to be a deeper explanation for why this method works. It would be better if the authors could provide more insights.
How do the degree of cropping and resizing, as well as the attention weights, affect the results? I don’t seem to see any related discussion or analysis on this.

- How do the degree of cropping and resizing, as well as the attention weights, affect the results? I don’t seem to see any related discussion or analysis on this.

**Questions:**

- Was there any weighting operation applied during the ensemble process?
- Can the authors discuss the relationship between this method and model uncertainty? When we perform multiple forward passes on the model and take the average, we are essentially obtaining model uncertainty. Can this method be understood as adding more logical constraints, but essentially optimizing the answer through model uncertainty?
- Besides logit ensemble, are there other levels of ensemble that could be attempted?

---

> ### Author Response · Authors · 2024-11-17
> **Response to Reviewer mCVa (1/2)**
>
> We appreciate the constructive feedback and the insightful questions, which provide valuable directions to further enhance the clarity and depth of our work.
>
> ---
>
> ## **Weaknesses**
>
> As demonstrated in our pilot study, sub-images generally contain fewer objects and maintain a relatively higher resolution for each object compared to original images. This enables the model to better focus on relevant objects to text query, which is critical for mitigating object hallucinations and improving overall performance. Figure 6 further demonstrates that our method allocates higher attention to relevant regions, effectively mitigating hallucination compared to existing approaches. Additionally, various benchmarks on hallucination and description tasks validate the effectiveness of our approach.
>
> In Appendix B, we outline the rationale behind the degrees of cropping and resizing employed in the method. The crop size of $336 \times 336$ is chosen to match the input image size of the vision encoder (CLIP-ViT-L-336px [1]). Since most images in the experiments range from 400 to 600 in width and height, each image is split into four parts to balance the trade-off between capturing sufficient detail and avoiding excessive fragmentation. Further splitting could risk cutting objects or reducing the size of regions, which might negatively affect the results. Using a fixed crop size to divide an image allows for overlaps between sub-images, ensuring that objects near the boundaries are not improperly split. To evaluate the effectiveness of crop size and overlap, we compare our method with a baseline approach where images are divided into four equal quarters without overlap. As shown in Table R1, incorporating overlaps effectively prevents improper object splitting and significantly improves performance.
>
> **Table R1. Results on the POPE COCO benchmark with and without overlap between sub-images.**
> | Setting       | Method                        | F1 Score | Accuracy |
> |---------------|-------------------------------|----------|----------|
> | **Random**    | ED (w/o overlap) |  84.68  | 86.27    |
> |               | ED                             | **88.84**    | **89.63**    |
> | **Popular**   | ED (w/o overlap) |  83.47  | 84.97    |
> |               | ED                             | **87.34**    | **88.03**   |
> | **Adversarial** | ED (w/o overlap) | 80.57    | 81.40    |
> |               | ED                             | **84.59**    | **84.97**    |
>
>
> To analyze the role of attention-guided weights, we conduct additional experiments where all sub-images are assigned uniform weights. Even under these conditions, as shown in Table R2, ED demonstrates consistent performance improvements compared to a method using uniform weights, further substantiating its robustness.
>
> **Table R2. Results on the POPE COCO benchmark with and without attention-guided weight.**
> | Setting       | Method                        | F1 Score | Accuracy |
> |---------------|-------------------------------|----------|----------|
> | **Random**    | ED (w/o attention-guided weight) | 80.68    | 83.60     |
> |               | ED                             | **88.84**    | **89.63**    |
> | **Popular**   | ED (w/o attention-guided weight) | 80.02    | 82.90     |
> |               | ED                             | **87.34**    | **88.03**   |
> | **Adversarial** | ED (w/o attention-guided weight) | 78.44    | 81.23    |
> |               | ED                             | **84.59**    | **84.97**    |
>
> ---

---

> ### Author Response · Authors · 2024-11-17
> **Response to Reviewer mCVa (2/2)**
>
> ## **Questions**
>
> ### **2.1. Weighting in the Ensemble Process:**
>
> As described in Equation 5 of the main text, the logits are weighted using their corresponding attention-guided weights, which are specifically designed to reflect the importance of each sub-image. Table R1 shows that the attention-guided weights consistently outperform uniform weights, achieving higher F1 Scores and Accuracy across all settings. These results clearly demonstrate the effectiveness of the proposed method.
>
> ---
>
> ### **2.2. Interpretation of the Relationship Between the Method and Model Uncertainty:**
>
> Our method can be interpreted as an advanced approach to addressing model uncertainty by incorporating logical constraints rather than relying solely on simple averaging. The logits derived from sub-images reduce the influence of unnecessary objects in high-resolution regions, resulting in more refined and contextually relevant representations. Attention-guided weights prioritize key features, enhancing the reliability of ensemble outputs. This approach mitigates hallucination while aligning with uncertainty reduction principles, optimizing confidence and performance.
>
>
> ---
>
> ### **2.3. Other Levels of Ensemble:**
>
> We have not explored other ensemble methods in this study, but a potential approach could involve treating each sub-image independently, performing inference on each, and combining the resulting outputs using an off-the-shelf model. Further refinement and optimization of this agentic ensemble approach could serve as a promising direction for future research.
>
> ---
>
> [1] Radford, A., Kim, J. W., Hallacy, C., Ramesh, A., Goh, G., Agarwal, S., ... & Sutskever, I. (2021, July). Learning transferable visual models from natural language supervision. In *International Conference on Machine Learning* (pp. 8748-8763). PMLR.

---

> > ### Author Response · Authors · 2024-11-28
> >
> > Dear Reviewer mCVa,
> >
> > We sincerely appreciate your valuable feedback on our paper. In response, we have carefully prepared a detailed rebuttal to address the points you raised. With the author-reviewer discussion period nearing its end, we are happy to clarify any remaining questions or discuss any further concerns you may have, if time allows. If our responses have resolved your concerns, we would be grateful to hear your thoughts.
> >
> > Thank you for your time and thoughtful consideration.

---

### Official Review · Reviewer_2dSM · 2024-11-09

**Soundness:** 3
**Presentation:** 3
**Contribution:** 3
**Rating:** 6
**Confidence:** 5

**Summary:**

This paper proposes a new Ensemble Decoding (ED) strategy to mitigate the object hallucination issue with Large Vision-Language Models (LVLMs). The proposed approach is motivated by the hypothesize that irrelevant objects and low object resolution in images are likely to impact performance negatively. In ED, the input image is split into sub-images and combines logit distributions by assigning weights through the attention map. ED adaptive plausibility constraint is proposed to calibrate logit distribution. The FastED, an optimized variant of ED, balances performance with speed by selecting a sub-image with the highest mean attention score from the original image. ED shows improved performance in several benchmarks

**Strengths:**

1. The ensemble decoding strategy proposed in this work is well-motivated and sound by zooming into local regions which is more important for decoding at each decoding step. The proposed method also does not require any training.
2. Starting from the baseline model, several optimization techniques have been proposed to improve model accuracy and running efficiency, including adaptive plausibility constraint to only keep the most plausible tokens, and a fast version ED to only focus on one particular sub-region in the image.
3. Strong performance and new state-of-the-arts have been reported on several benchmarks.

**Weaknesses:**

The proposed ensemble encoding strategy, fuses the logits extracted from the original image and multiple local crops. This method makes sense, but introduces much more computation burden for real-world deployment. The FastED only processes one local crop and have some accuracy drops. However, one simpler alternative is to move the ensemble to the input, and concat the multiple images in the prompt. We can imagine that this method is much faster than ED/FastED because only one feed-forward is needed. There is no discussion about where to do the ensemble.

**Questions:**

1. it is unclear how to get the $d\times d$ attention matrix in L161, Section 3.1, where d is the number of image patches. Can the author elaborate the dimension of $Q_t$ and $K_t$ in detail?
2. As for the logit ensemble, the generated logit should has the same of the size of LLM tokenzier? It is unclear why the proposed adaptive plausibility constraint improves performance by only keeping the elements in #p_ED# which are larger than a threshold?
3. Do we have any assumption that the length of the generated tokens have the same length? How should we do the logit ensemble when we have different number of generated tokens for different images.
4. Even though the proposed method is training-free, it also introduces a lot of hyper-parameters $\alpha$, $\beta$, $N$, $H$ and $K$. how to tune these hyper-parameters is unclear.

---

> ### Author Response · Authors · 2024-11-17
> **Response to Reviewer 2dSM**
>
> We appreciate your valuable feedback regarding our method and its relation to real-world deployment. Your thoughtful suggestions have inspired further reflection on our approach.
>
> ## **Weaknesses**
> We acknowledge that our proposed ensemble decoding strategy introduces computational overhead, and we agree that it is not yet the optimal approach in terms of inference speed. Your suggestion of using a multi-image input strategy to process multiple images in a single feed-forward computation is indeed an interesting and innovative idea. This method could reduce computational costs and is worth further exploration.
>
> However, implementing such an approach would require Vision-Language Models (VLMs) that are specifically trained to handle multiple images as inputs for generating text responses. Unfortunately, most existing VLMs, such as LLaVA 1.5 and BLIP-2, are designed and pre-trained to accept only a single image as input. Consequently, these models do not currently support multi-image input within a single forwarding step, limiting their applicability in this context.
>
> For real-world deployment, we believe that a balanced approach is necessary, taking into account both sensitivity to hallucination issues and inference speed. Depending on the use case, different strategies such as ED, FastED, or Regular decoding might be more suitable.
>
> ---
> ## **Questions**
> ### **1.1. Dimension of $d$, $Q_t$, and $K_t$:**
>
> In Section 3.1 (L161), $d \times d$ refers to the number of patches used by the image encoder of the 2D VLM. It depends on the VLM employed. For example, we used LLaVA 1.5 with a CLIP-ViT-L-336px [1] image encoder, which has $24 \times 24$ patches.
>
> Regarding $Q_t$ and $K_t$, their dimensions follow the standard self-attention mechanism used in Transformer. Specifically, $Q_t$ is the query matrix at time $t$, with dimensions ($1 \times \frac{\text{hidden dimension}}{\text{number of heads}}$). $K_t$ represents the keys from time 1 to $t$, with dimensions ($\text{sequence length} \times \frac{\text{hidden dimension}}{\text{number of heads}}$). To improve clarity, we will revise $K_t$ to $K_{\leq t}$ in the manuscript.
>
> ---
>
> ### **1.2. Logit Ensemble and Adaptive Plausibility Constraint (APC):**
>
> In our ED method, the logit ensemble is computed after forwarding $n$ split images individually through the same model. The final ensembled logits share the same vocabulary size as the LLM tokenizer.
>
> The proposed ED APC emphasizes sub-image logits by filtering out logits below a threshold, ensuring such tokens are not amplified during the ensemble. It reflects our intuition that sub-images often contain richer details. As shown in Figure 7, ED APC consistently outperforms the original image-focused general APC used in prior work.
>
> ---
>
> ### **1.3. Handling Different Token Lengths:**
>
> ED does not assume or constrain the length of generated tokens. Instead, logit distributions from sub-images are ensembled dynamically at each decoding step (as illustrated in Figure 3). If the end token is chosen in the ensembled logits, the decoding process terminates, similar to other LLMs/VLMs.
>
> ---
>
> ### **1.4. Hyperparameter Tuning:**
>
> Hyperparameters $\alpha$ (in logit ensemble) and $\beta$ (in ED APC) determine the emphasis on sub-images, with larger values favoring sub-image contributions. As shown in Table 4, when ED APC is applied, ED demonstrates robust performance across varying $\alpha$ values. To further explore the impact of $\alpha$, we conduct additional experiments. Specifically, we perform ablation studies on $\alpha$ in scenarios where ED APC is not used  ($\beta=0$). The results, as summarized in Table R1, indicate that larger $\alpha$ values consistently lead to improved performance. This suggests that emphasizing sub-images has a positive effect on overall model performance.
>
> **Table R1. Results on the POPE COCO benchmark with varying $\alpha$ values and without ED APC.**
> | α | F1 Score | Accuracy |
> |--------------------|----------|----------|
> | 0 (Normal Decoding)| 79.67 | 81.40 |
> | 0.3 | 80.27 | 81.80 |
> | 0.5 | 80.92 | 82.22 |
> | 0.7 | **81.66** | **82.73** |
>
>
>
> For $H$ and $K$, used in computing attention scores, we adopted default settings from prior work [2]. Exploring optimal values for these parameters is beyond the scope of this study but could be a potential avenue for future research.
>
> ---
>
>
> [1] Radford, A., Kim, J. W., Hallacy, C., Ramesh, A., Goh, G., Agarwal, S., ... & Sutskever, I. (2021, July). Learning transferable visual models from natural language supervision. In *International Conference on Machine Learning* (pp. 8748-8763). PMLR.
>
> [2] Lee, S., Park, S. H., Jo, Y., & Seo, M. (2023). Volcano: mitigating multimodal hallucination through self-feedback guided revision. *arXiv preprint arXiv:2311.07362.*

---

> ### Author Response · Authors · 2024-11-28
>
> Dear Reviewer 2dSM,
>
> Thank you for your insightful comments on our paper. We have carefully prepared a rebuttal, making every effort to address your concerns. As the author-reviewer discussion period approaches its conclusion, we are more than willing to respond to any unresolved issues or further questions you might have, should time permit. If our rebuttal has satisfactorily addressed your concerns, we would greatly appreciate hearing your thoughts. Please feel free to reach out with any additional questions or feedback regarding our paper.
>
> Thank you for your time and consideration.

---

### Meta-Review · Area_Chair_CNSR · 2024-12-18

**Metareview:**

The paper tackles the challenge of object hallucination in modern vision-language models. The authors propose an ensemble decoding strategy (ED) that feeds high resolution chunks of the input image to the model and decodes these chunks independently. These "ensemble" predictions are then combined together by using attention-based weights. The experiments in this work show that ED reduces hallucinations on POPE and CHAIR benchmarks.

Strengths
1. ED as a strategy is well motivated by the fact that image resolution is critical for correctly identifying details about an image
2. The FastED variant is also practically useful by trading off accuracy for speed
3. The paper is well presented and experiments are explained well
4. ED can be used for calibrating logit distributions

Weaknesses
1. The paper trades off inference compute for accuracy. While this is useful, it does create practical issues.
2. I found the explanation for the attention based logit re-weighting hard to understand. It would be good to make this clearer.
3. The experiments are limited to a few datasets which reduces the impact of the approach. It would have been good to show that ED is more generally effective for object recognition.

Justification
This paper studies an important problem, is technically correct, and has convincing results. The paper also unanimously received positive scores.

**Additional Comments On Reviewer Discussion:**

The reviewer concerns were addressed by the authors and all the reviewers remain positive about the work.

---

### Decision · Program_Chairs · 2025-01-22

Accept (Poster)